# A Study on the Life Estimation and Cavity Surface Degradation Due to Partial Discharges in Spherical Cavities within Solid Polymeric Dielectrics Using a Simulation Based Approach

**DOI:** 10.3390/polym13030324

**Published:** 2021-01-20

**Authors:** Johnatan M. Rodríguez-Serna, Ricardo Albarracín-Sánchez

**Affiliations:** 1Departamento de Ingeniería Eléctrica, Electrónica, Automática y Física Aplicada, Escuela Técnica Superior de Ingeniería y Diseño Industrial (ETSIDI), Universidad Politécnica de Madrid (UPM), Ronda de Valencia 3, 28012 Madrid, Spain; ricardo.albarracin@upm.es; 2Departamento de Ingeniería Eléctrica, Facultad de Ingeniería, Universidad de Antioquia (UdeA), Calle 67 No. 53-108, 050010 Medellín, Colombia

**Keywords:** partial discharges, time-to-breakdown, polymer degradation, prognosis, dielectric breakdown

## Abstract

Partial Discharges (PD) in cavities are responsible for the greatest ageing rate in polymeric solid dielectrics due to chemical and physical deterioration mechanisms activated by the charge carriers, Ultra Violet (UV) radiation and local temperature rising during PDs activity. From the above, it is necessary to develop prognosis tools based on PDs measurements as diagnostic quantities in order to infer the time-to-breakdown, life, of solid dielectrics for improving the reliability of electrical assets, especially in current applications where they are subject to great electrical stresses in voltage frequency and magnitude. In this paper, the degradation in polymeric materials induced by PDs in cavities is briefly discussed from a phenomenological point of view, and then it is quantitatively evaluated using a simulation-based approach and a new proposed damage function. The time-to-breakdown calculated from simulations exhibits good agreement when compared with experimental measurements. Additionally, an analysis on the effect of the magnitude and frequency of the applied voltage on the degradation rate is also presented and the effectiveness of a degradation indicator, proposed by other authors, is evaluated under different stress conditions.

## 1. Introduction

In order to improve the maintenance actions on electrical assets, it is necessary to make an adequate diagnosis of the equipment health using measurements and indexes that allow to evaluate the actual operating conditions of the equipment [1]. For insulation systems and dielectric materials, phenomenological macroscopic life models allow to infer the time-to-breakdown using extrapolation from accelerated tests results [2]. However, local defects as cavities, protrusions or impurities, induce the acceleration of ageing rates, which makes difficult the precise and reliable life estimation under different stress conditions [3].

Polymeric materials, as well as polymerizable resins, are used in the different subsystems of the electrical assets insulation systems because of its good dielectric properties, elasticity, thermal stability and resistance against moisture, dirt and chemical contamination [4]. In addition, they allow to increase the reliability, cost effectiveness and power density of electrical apparatus, e.g., electrical machines, in modern applications such as mobility, aerospace and high speed [5,6,7]. 

Cavities can appear in polymeric materials as a consequence of the manufacturing and curing processes [8], and Partial Discharges (PD) activity can be started inside them, accelerating the ageing process of the dielectric materials [9]. During PDs in cavities, electrons, and also ions, can acquire high-kinetic energies due to the increased fields and long-free paths [10]. These highly energetic electrons collide with the gas–solid interface and can cause the dielectric degradation via impact ionization, C–H bond scissions and chemical reactions induced by free radicals [11,12]. The latter chemical processes, along with oxidation, are the most obvious mechanisms for polymer degradation driven by PDs in cavities [9,13]. 

As PDs activate the fastest degradation mechanism in polymeric solid dielectrics due to chemical and physical deterioration mechanisms activated by the charge carriers, UV radiation and local temperature rising during PDs activity [8,14], their measurable magnitudes, i.e., partial discharge (PD) charge, PD energy, PD pulse shape, etc., can be used as effective diagnostic quantities, specially, in organic dielectric materials or oil-paper [15,16]. In addition, the types of defects [17], as well as some of their characteristics, such as, shape, diameter and ageing conditions, among others, can be inferred from PD measurements [18,19]. However, an accurate prediction of the remaining useful life of the equipment from PD measurements is still not available.

On the other hand, nowadays applications such as hybrid AC and DC transmission schemes and nonlinear loads impose stresses conditions different to those considered at rated conditions. In hybrid AC and DC systems, dynamic ageing and life modeling is required because ageing factors and mechanisms are changing depending on power planning and dispatch constrains [20]. Under DC voltage applied, the field distribution is different from the case of AC voltage, which causes that the inception of PDs activity at internal cavities is variable depending on their location and the type of applied voltage, AC or DC. In addition, at DC, the PD inception voltage can become lower than under AC because the conductivity increases due to the temperature rising by load increments [20]. Harmonic components are also able to increase the thermal and electrical stresses due to superposition with the fundamental component causing increments in the PD rate, the PD pulse magnitude and, consequently, the degradation rate [21]. In medium-voltage rated motors, the PD pulse magnitude and PD rate will increase under an inverter supply, however longer life times are obtained when both, the number of inverter voltage levels and the impulse rise time, increase, with a maximum when the voltage corresponds to a sinusoidal wave shape [22]. 

In this paper, it is presented a study on the degradation of polymeric materials induced by PDs in spherical cavities inside solid dielectrics implementing an innovative simulation based approach that uses a novel damage function and allows to quantitatively inferring the time-to-breakdown of the insulation system. Additionally, based on simulation results, the effectiveness as well as the period of predominance above each other of the chemical and physical degradation mechanisms are discussed. On the other hand, the simulation-based approach is used for numerically evaluating the effect of the applied voltage frequency and magnitude on the degradation rate and analytical expressions for the trend lines of the accumulated damage or ageing as a function of voltage magnitude and frequency were determined.

In this study it is considered that the ageing of the solid dielectric material is driven mainly by PDs in cavities which is supported by the fact that the breakdown is determined by the most harmful condition and the greatest ageing rate [13]. The degradation induced by PDs in spherical cavities at different stages, phases, during ageing is estimated through a method that uses a microscopic life model inspired in the Serra et al. work [23], in which the polymer C–H bonds are broken via Dissociation by Electron Attachment (DEA). DEA is a chemical process where electrons are temporarily trapped in a resonant state of the molecules causing its fragmentation [24]. The implemented method makes it possible to infer quantitatively the time-to-breakdown under the electrical stresses imposed at each phase of the aging process. It is assumed that the ageing phases can be clearly differentiated by the variations in the Phase-Resolved PD (PRPD) patterns that have already been analyzed and characterized by other researchers in [25]. On the other hand, the effects of applied voltage magnitude and frequency on the ageing rate are analyzed using simulation results. The effect of the space charge and physical mechanisms on the ageing rate are also discussed and an ageing rate indicator, proposed by other authors in [26], is also evaluated under different stress conditions. Statistical analyses are included for considering the stochastic characteristics of the PD behaviour.

The paper is organized as follows: first, the materials and methods employed for the case studies are summarized in Section 2; then, the degradation induced by PDs in polymeric materials is briefly discussed in Section 3. The method for calculating the degradation induced by PDs and for estimating the life is presented afterwards in Section 4; next, the simulation results for the considered case studies are presented and discussed in Section 5. Finally, some conclusions are depicted in Section 6.

## 2. Materials and Methods

For this work, three case studies were implemented. The first case study allows evaluating the degradation along different ageing phases differentiated by the variations in the PRPD patterns. The time-to-breakdown from unaged conditions is calculated using the life estimated at each ageing phase. Then, numerical results are compared with measurement observations, which allows showing the effectiveness of the implemented method for estimating the life of solid dielectrics subject to PDs in cavities using simulations. On the other hand, the second and third case studies allow analyzing, respectively, the effect on the degradation rate of the applied voltage magnitude and frequency. 

The first case study was presented in [25], while the second and third were presented, respectively, in [27,28]. The sample preparation is described in each corresponding reference. The Table 1, summarizes the materials description for each case study as well as the characteristics of the test arrangements. 

The test arrangement for all the case studies corresponds to a spherical cavity immersed in a solid dielectric bulk, which is put into a flat electrode configuration with a dielectric separation D (m), see Table 1. An AC voltage source is applied to the upper electrode while the lower is grounded. 

The parameters of media for the first case study at each ageing phase were presented in [29]. It was assumed that during aging the ionization characteristics of the gas in the cavity are similar to those of air. This approximation is reasonable in the sense that the gaseous decomposition and the products of the reactions (such as CO_2_, CH_4_, and NOx) are not very different from air in their ionization characteristics [25]. Similarly, the parameters of media for the second and third case studies are presented, respectively, in [27,28]. 

The method for calculating the induced damage and the time-to-breakdown due to PDs in spherical cavities inside solid dielectrics consists of the following two steps:PDs simulation; andPD induced damage calculation.

At the first step, PDs are simulated for each case study using the hybrid PD-Finite Element Analysis model presented in [29]. The following assumptions were made for the simulations and calculations:Discharges propagate along the symmetry axis and are streamer-like [30].The PD deploys charge carriers of opposite sign at the cavity surface where all they are initially concentrated at the intersection point between the symmetry axis and the cavity surface. After that, the charge assumes a field-dependent distribution [29].As the damage depends on the charge carrier collision and they are initially concentrated on the intersection point between the symmetry axis and the cavity surface, where the streamer impacts with the cavity surface, it is considered that the damage accumulates at that point [19].

In comparison to Townsend-like, streamer-like PDs cause severe degradation to polymers which allows to conclude that cavities with dimensions < 10 μm will not be eroded by PDs [10]. Streamer-like PDs produce local degradation because they inject charge and produce oxidation locally, generating semiconducting patches in the colliding area [8]. 

At the second step, the degradation induced by PDs in the solid dielectric, polymer C–H bonds broken via DEA [23], is calculated as a function of the PD-electrons energy distribution and the scattering rates in the polymer, this is detailed in Section 4. As the electrical tree propagation is faster than its inception [31], the time-to-breakdown, life, is considered as the time required to the formation of a damaged zone of size large enough for sustain PDs independent from the PDs in the cavity and the tree propagation can be incepted. 

## 3. Degradation of Polymeric Material Induced by PDs in Cavities

The continuous PD activity induces the formation of pits and craters on the cavity surface due to the erosion of the dielectric material through the following physical and chemical mechanisms [8,32,33]:Bombardment of the cavity surface by charge carriers and photons;Increment on the local temperature by high temperature discharge gases; andChemical reactions activated by excited molecules or chemical compound in the gas and solid phase, particularly oxygen and ozone.

Those pits and craters can lead to the propagation of electrical trees due to local field enhancements [3,34]. In oxygen free conditions, the continuous ion bombardment of the surface is able to produce craters or pits. However, due to their low energy, is required the simultaneous occurrence of great temperature risings and under normal conditions it only appears as a worsening factor of existing damage [8]. On the other hand, the increment on the local temperature in the cavity can accelerates the chemical reactions, but other thermal degradation mechanisms, such as, the direct melting are negligible [35].

PDs in cavities inside solid dielectrics are cause and consequence of material ageing and the effectiveness of the aforementioned mechanisms on the material degradation, depends on the gas and solid composition, temperature, gas pressure, cavity surface conductivity, and roughness, as well as the energy dissipated by the PDs [36]. In turn, the cavity gas and surface characteristics influences the PD behaviour because the gas ionization parameters are function of its composition, and the charge drift or recombination and emission rates at the cavity surface depend on its conductivity and roughness [26]. In oxygen rich conditions appear large PD pulses, which erode uniformly the cavity surface by ion bombardment and oxidation. On the other hand, in nitrogen and moisture conditions appear swarming micro PD pulses that produce localized degradation due to ion bombardment. In air, the PD degradation exhibits the combined effect [32].

Chemical reactions activated by hot electrons determine the rates of oxygen consumption, in gas and solid phases, and by-products generation, which modifies the cavity gas composition and pressure, as well as the cavity surface roughness. At the initial ageing phases, in oxygen rich conditions, the PD energy allows creating active oxygen species (O, O_2_, and O_3_) which degrade uniformly the cavity surface and a synergy may exist between DEA and autoxidation [37]. The dissociation of C–H bonds produces H^−^ ions and polymer free radicals, R˙ which are formed after an initiation step X, caused by oxidation, UV absorption, electron collisions or ionizing radiation as [8]:(1)$$X+R_{a+b}\to R_{a}^{*}+R_{b}^{*}$$

Those polymer free radicals are very reactive and promote the chain scission and cross-linking through chain reactions, especially in oxygen rich conditions, that finish when all the energy has been consumed or an antioxidant reaction occurs. The activation energy for autoxidation is low and this is the reason why this process is very important in the tree inception under AC voltage [12].

During ageing, an oxidized layer containing carbonyl radicals (=CO) appear at the cavity surface, which modifies its conductive characteristics, affecting indeed the PD inception and extinction magnitude, the electric field distribution and streamer landing pattern [32]. The non-uniform deposition of by-products cause that some areas are prone to experience intensive bombardment [38]. In epoxy specimens, the oxidation process is dependent on the curing process because it has been shown that when anhydride hardener is used it reacts with most of the oxygen leaving an atmosphere that is 80–98% nitrogen. Nevertheless, in air and nitrogen the PD behaviour is quite similar [39], which allows to infer that chemical reactions in the surface are the most important and that the oxygen in the gas or in the polymeric matrix participates in the surface oxidation. 

After experimental observations of PDs that induced degradation in spherical cavities enclosed in epoxy resin [40], it have been found that despite some by-products were encountered at the equatorial zones perpendicular to the electric field in the cavity, the surface degradation is severe at the polar zones closer to the High-Voltage (HV) electrodes. At those polar zones, semiconducting crystals made of oxalic acid were found, which due to field enhancement, concentrate the PD activity in their vicinity, increasing pitting and originating channels from the void surface. From that observation, it was concluded that crystals formation is in fact the first stage in the tree inception [40]. As the crystals modify the cavity geometry and the PDs inception location, the appearance of crystals can be detected analyzing the asymmetries in the PRPD pattern, however the appearance of the channels cannot be easily detected because the low magnitude of the PD pulses in them [41]. 

The cavity gas pressure diminishes due to oxygen consumption in chemical reactions, in the gas and the cavity surface, and diffusion of gases into the dielectric bulk [42]. Both, the cavity surface conductivity and cavity gas pressure determines the PD inception voltage and rate and are considered as the main parameters that affect tree initiation [43]. 

On the other hand, due to the aforementioned changes on the cavity surface and gas composition and pressure, the PD behaviour along the ageing period changes. This means that the ageing rate changes, and probably the degradation mechanism too, and for this reason the ageing and life models have to clearly specify the ageing phase for which it applies [44]. In this study, it is considered that the chemical mechanism is the predominant and it is active along all the ageing phases. 

Under practical conditions, the changes in the gas and surface conditions, previously mentioned, can be inferred from the PD behaviour and PRPD patterns, analyzing the changes in the PD rate, pulse shape and magnitude, time lag, etc., which can be used as diagnostic indicators. However, a precise and meaningful diagnostic of the dielectric and life estimation based on PD measurements is impossible without analyzing the PD phenomenon at a microscopic scale from a physical and chemical point of view [45]. In addition, the overall ageing process is determined by thermal, electrical, environmental and mechanical stresses and the stochastic behaviour of PDs, requires the estimation of life from a statistical framework [46].

Microscopic life models provide a physical explanation to the ageing process and fittings of parameters relative to media or conditions of tests, as in the phenomenological macroscopic life models, are not required [47]. Furthermore, microscopic life models can give direct comprehensive relationships between the physical, chemical and microstructural characteristics of polymers and the degradation mechanisms associated with PDs in local defects [48]. In addition, microscopic life models with measurements can be used as prognosis tools [49,50]. 

## 4. Calculation of the Degradation Induced by PDs and Life Estimation Approach

Taking into account theoretical and experimental studies [38,51], it is assumed that PDs in spherical cavities can incept treeing, these being the last stage of the aging process before breakdown. Once the electrical tree is incepted, i.e., the first tree channel has a length enough for develop self-sustained PDs independent from those at the cavity, electrical tree continues propagating via a physical degradation mechanism [31,41]. On the other hand, chemical degradation due to PD in tree channels tend to increase the channel diameter [10].

As it was described in the Section 3 of this document, both, physical and chemical mechanisms are able to produce chain-scissions. The simulation-based approach implemented in this study limits the scope to consider only the chemical mechanism for the different ageing phases. This chemical mechanism also induces irregular morphological changes, as the formation of crystals that cannot be simulated using the approach proposed here. Experimental measurements, should allow inferring the crystals growth rate and location, in order to improve the modeling and simulation of the cavity geometry and their effect on the PD inception voltage and rate and the streamer landing location, which indeed affects the most damaged area and the time to reach the critical length for starting the self-sustained tree propagation.

### 4.1. Calculating the PD Induced Damage by DEA

Highly energetic electrons, hot electrons, accelerated by the local electric field during PDs collide with the cavity surface and release their energy upon a region close to the collision point, and if the scattering rates of the polymer are known [52], the damage, assumed as the number of broken C–H bonds through a DEA process, can be estimated. It is also assumed that hot electrons in the solid dielectric are far below critical conditions for breakdown. i.e., the electric field strength magnitude at the solid bulk is lower than in the cavity gas and the dielectric breakdown strength magnitude is higher in the solid than in the gas and undergo a fast thermalization by impact ionization and DEA of C–H bonds. DEA is responsible for chemical damage that irreversibly accumulates in the polymer because of consecutive avalanches over time. All the thermalization process occurs in a slab of thickness $D_{dis}\sim 400$ Å [13], for this reason, the damage induced by hot electrons must be confined to this slab.

The method of analysis implemented in this study is based on the microscopic model proposed by Serra et al. [23]. The PD activity is simulated using the hybrid PD-Finite Element Analysis model presented in [29]. The number of electrons generated during the *i*-th PD is calculated as:(2)$$N_{el}^{i}=\frac{q_{PD}^{i}}{q_{0}},$$
where $q_{PD}^{i}$ (C) is the PD charge and $q_{0}$ (C) is the elementary charge. The rate of electrons colliding with the polymer surface during the *i*-th PD corresponds to:(3)$$R_{el}^{i}=\frac{N_{el}^{i}}{\Delta t_{PD}^{i}},$$
where $\Delta t_{PD}^{i}$ (s) is the duration of the *i*-th PD. According to the model in [23], the time to dissociate by DEA the half of the C–H bonds, $N_{CH}$, in the volume of a disrupting slab of size enough for starting an electrical tree can be calculated as:(4)$$t_{dis}=\frac{N_{CH}}{2R_{el}F_{hot}F_{eff}},$$
where $F_{hot}$ and $F_{eff}$ are, respectively, the fraction of hot electrons produced during the PD and the fraction of collisions effective for DEA. From Equation (4), the following damage function can be defined:(5)$$f_{PD}^{i}=\frac{2R_{el}^{i}F_{hot}F_{eff}}{N_{CH}}\Delta t_{PD}^{i},$$
where $f_{PD}^{i}$ is the damage accumulated, as a portion of broken C–H bonds in a cylindrical slab of length $D_{dis}=400$ Å and circular cross section of $S_{c}=\pi {\left(1\right)}^{2}$ [μm^2^] [53], at the cavity surface due to the *i*-th PD. The damage accumulates until $\sum f_{PD}^{i}=1$ when $t=t_{dis}$ (s). The damage continues accumulating up to the deteriorated region reaches a critical length, $d_{c}$ (m), from which an electrical tree can be started and propagates independently from the PDs at the cavity, $d_{c}=3$ μm [13]. The life, L (s), can be calculated as:(6)$$L=\frac{d_{c}t_{sim}}{D_{dis}\sum_{t_{sim}}f_{PD}^{i}},$$
where $\sum_{t_{sim}}f_{PD}^{i}$ is the accumulated damage during the simulation time $t_{sim}$ (s). For determining $F_{hot}$ the following approach is used: first, the electron energy distribution function is determined as a function of the electric field strength magnitude in the cavity, and then the fraction of hot electrons, electrons with energy ≥8 eV, is calculated. The electron energy distribution function is calculated using the Maxwellian shape [54]:(7)$$f\left(w,E_{cav}\right)={\underline{w}}^{-3/2}w^{1/2}\beta_{1}\exp\left(-\frac{w\beta_{2}}{\underline{w}}\right),$$
where w (eV) and $\underline{w}$ (eV) are, respectively, the electron energy and mean electron energy, $\beta_{1}=\Gamma {\left(5/2\right)}^{3/2}\Gamma {\left(3/2\right)}^{-5/2}$, $\beta_{2}=\Gamma \left(5/2\right)\Gamma {\left(3/2\right)}^{-1}$ and Γ is the Gamma function. The mean electron energy is calculated as:(8)$$\underline{w}=\underline{\lambda_{e}}q_{0}E_{cav},$$
where $\underline{\lambda_{e}}\left(p,T\right)$ (m) is the mean free path of electrons in a gas, for air $\underline{\lambda_{e}}=3.792\times 10^{-7}$ m at $p=p_{0}=100$ kPa and $T=T_{0}=273.15$ K [55], and $E_{cav}$ (V·m^−1^) is the electric field strength magnitude at the cavity centre. $F_{hot}$ can be calculated as a function of the electric field strength magnitude in the cavity as [13]:(9)$$F_{hot}=\frac{\int_{w=8\mathrm{eV}}^{\infty }f\left(w,E_{cav}\right)dw}{\int_{0}^{\infty }f\left(w,E_{cav}\right)dw}.$$

For determining the fraction of hot electrons effective in producing DEA, $F_{eff}$, it is necessary to solve the Boltzmann equation in the energy-time domain in the solid dielectric [56]:(10)$$\frac{\partial n\left(w;E_{cav},t\right)}{\partial t}+\frac{\partial F\left(w;E_{cav},t\right)}{\partial w}=G,$$
where $n\left(w;E_{cav},t\right)$ [m^−3^] is the volume density of free electrons with energy in the range [w,w+dw], $F\left(w,E_{cav},t\right)$ (W·m^−3^) is the electron power density and G (m^−3^·s^−1^) is the factor accounting for the electron generation or removal through inelastic impacts within the dielectric media. Alternatively, considering that $F_{eff}$ depends on the electron scattering parameters of the solid dielectric and on the energy distribution of electrons impinging the surface, which is independent on the cavity diameter, $F_{eff}$ can be calculated from the curves reported in [13], assuming that $F_{eff}\approx 0.2$ for $E_{cav}\geq 60$ kV·mm^−1^. Using the least squares method, the following analytical expression was determined:(11)$$F_{eff}=0.0574\ln\left(E_{cav}\right)-0.0472,$$

$E_{cav}$ is in kV·mm^−1^ and the coefficient of determination obtained was $R^{2}=0.981$. It is assumed that the scattering parameters of epoxy are similar to that of polyethylene. For considering, that during each PD event the electric field strength magnitude inside the cavity is quenched by the field produced by the PD charge, $F_{hot}$ and $F_{eff}$ are calculated for the *i*-th PD event as:(12)$$F_{hot}^{i}=\frac{1}{E_{cav}\left(t_{PD}^{i}\right)-E_{Ext}}\int_{E_{Ext}}^{E_{cav}\left(t_{PD}^{i}\right)}F_{hot}\left(E_{cav}\right)dE_{cav},$$
(13)$$F_{eff}^{i}=\frac{1}{E_{cav}\left(t_{PD}^{i}\right)-E_{Ext}}\int_{E_{Ext}}^{E_{cav}\left(t_{PD}^{i}\right)}F_{eff}\left(E_{cav}\right)dE_{cav},$$
where $E_{cav}\left(t_{PD}^{i}\right)$ (V·m^−1^) and $E_{Ext}$ (V·m^−1^) are, respectively, the electric field strength magnitude in the cavity at the inception of the *i*-th PD and the extinction electric field strength magnitude.

### 4.2. Space Charge Effect and Physical Degradation Mechanism

According to the discharge avalanche model [57], once the void surface is sufficiently degraded, charges in the form of ionized gas can be injected through microscopic channels in the polymer and C–H bonds can be broken due to induced electron avalanches. The number of ionizations per discharge is calculated as [58]:(14)$$N_{D}^{i}=N_{0}^{i}\left[\exp\left(\alpha \left(E_{loc}^{i}\right)L_{b}\right)-1\right],$$
where $L_{b}=10$ μm is the unitary tree length channel [59], $N_{0}^{i}$ is the initial number of electrons available for starting the avalanches at the cavity surface, $E_{loc}^{i}$ (V·m^−1^) is the local electric field strength magnitude at the injection point and the ionization rate, $\alpha \left(E_{loc}^{i}\right)$, corresponds to [31,60]:(15)$$\alpha \left(E_{loc}^{i}\right)=\frac{1}{\lambda_{p}}\exp\left(-\frac{I_{p}}{q_{0}\lambda_{p}E_{loc}^{i}}\right),$$
where $I_{p}$ (eV), is the ionization energy and $\lambda_{p}$ (m) is the infinite field limit of the collision ionization path length. For polymers, they correspond, respectively, to 9.6 eV and 60 nm [60]. From Equation (15) a threshold field magnitude, $E_{th}$ (V·m^−1^), below which there is not charges injection in the micro-channels, can be defined as $E_{th}=I_{p}/q_{0}\lambda_{p}$ (V·m^−1^).

The charge left by PDs on the cavity surface increases the electric field strength magnitude in the solid dielectric, in points near the solid-gas interface, to values greater than the electric field strength at the centre of the cavity. The electric field strength magnitude produced by the surface charge density on the cavity surface at the distance r>a (m) from the centre of the cavity in the solid dielectric after a PD, can be calculated as [29]:(16)$$E_{ch}\left(r,\theta \right)=\frac{q_{PD}}{2\pi \varepsilon_{0}r^{2}}\sum_{n=0}^{\infty }\left[\frac{\left(4n+3\right)\left(2n+2\right)}{\left(2n+2\right)\varepsilon_{r}+\left(2n+1\right)}{\left(\frac{a}{r}\right)}^{2n+1}P_{2n+1}\left(\cos\theta \right)\right],$$
where $P_{2n+1}\left(\cos\theta \right)$ is the Legendre polynomial of cosθ of degree 2n+1, $\varepsilon_{r}$ is the relative permittivity of the solid dielectric, $\varepsilon_{0}$ (F·m^−1^) is the permittivity of vacuum and θ (rad) is the polar angle in spherical coordinates with origin at the cavity centre.

## 5. Results and Discussion

The geometry of the case studies is shown in the Figure 1. The specifications of the material, applied voltage and dimensions are presented in Table 1 for each case study.

It is considered that the damage induced by the hot electrons during each PD event depends on the polarity of the resultant electric field strength inside the cavity. Besides, it is accumulated at the centre of the anode on the cavity surface, which under AC fields, it alternates among the upper, blue, and lower, red, hemispheres at the inner cavity surface, respectively, *S*_1_ and *S*_2_ in Figure 1.

The hybrid PD-Finite Element Analysis model can be summarized as in the following steps [29]:The parameters of media, geometrical constants, boundaries and subdomains are defined;For each time-step the electric field strength is calculated using a Finite Element Analysis solver at all domains;For each time-step the inception criteria are electron existence and minimal electric field strength magnitude and both are verified;If the inception criteria are fulfilled, the cavity conductivity is increased and the time-step is diminished. The high-conductivity condition is maintained until the electric field strength magnitude is lower or equal to an extinction magnitude;If inception criteria are not fulfilled, the charge distribution on the cavity surface deployed by previous PDs is calculated as a function of the electric field strength inside the cavity, the time and the cavity surface conductivity;The induced charge is calculated evaluating boundary conditions at electrodes and parameters of media are reset for the following time-step. Calculations for the following time-step are executed as in step 2 until the required simulation time is reached.

In the hybrid PD-Finite Element Analysis model used here, as in the conductance and plasma models [30], the initiation and ending locations of the PDs are considered fixed specific points on the inner cavity surface, the centre of surfaces *S*_1_, *A*, and *S*_2_, *B*, Figure 1. This is because the electric field strength magnitude is the greatest at those points, which increases the probability of first electrons emission for incepting PDs [61]. Additionally, it is considered that once the PDs are incepted at points A or B, they propagate along the cavity symmetrical axis. This is because charges are accelerated by the electric field in the direction of the greatest electric field strength magnitude, until the opposite surface is reached at points B or A, where the charges left by PDs produce a field that opposes to the externally applied, quenching the PDs processes [62].

As the charges deployed by PDs are initially concentrated close to points A and B, Figure 1, where the streamer impinges the cavity surface and the electric field strength magnitude and the energy distribution of electrons are the greatest [14,61], it is assumed that the damage induced by hot electrons is accumulated at those points on the inner cavity surface. A precise evaluation of the real affected area requires the precise determination of the initiation and ending locations of PDs and the consequent damage accumulation at those locations. Plasma models can give a good alternative for modeling the precise PD landing pattern [30].

### 5.1. Case Study 1, PDs Induced Degradation at Different Ageing Phases

The first case study is the same than that considered in [25]. It corresponds to an air filled spherical cavity of radius a=1.25 mm in the middle of a linear, homogeneous and isotropic dielectric bulk of epoxy resin, see Table 1, of thickness D=3.5 mm, between two parallel plates. An AC 19.25 kV, 50 Hz voltage source is applied to the upper electrode while the lower is grounded. Simulations were implemented for five consecutive ageing phases, differentiated by the changes on the PD behaviour during the time under stress, as:Phase A, from unaged to 0.17 h;Phase B, from end of Phase A to 35.17 h;Phase C, from end of Phase B to 185.17 h;Phase D, from end of Phase C to 1235 h; andPhase E, from end of Phase D to 1285 h.

The parameters of media and the PD simulation model for each ageing phase are presented in [25,29]. The number of C–H bonds in the unitary slab is calculated as $N_{CH}=\rho_{CH}D_{dis}S_{c}$, where $\rho_{cH}$ (m^−3^) is the volume density of C–H bonds. Using the material parameters for epoxy resin presented in [63], the volume density of C–H bonds is calculated as 1.1507 × 10^29^ m^−3^.

At each ageing phase, thirty simulations were carried out during 500 cycles of the AC applied voltage. Figure 2 shows typical simulation results for the PRPD pattern as well as the accumulated damage during the simulation time.

In Figure 2 “Damage (A.U.)” is the magnitude of the accumulated damage at each separate ageing phase during the simulation time calculated with Equation (5). *S*_1_ and *S*_2_ correspond to the hemispherical surfaces described in Figure 1. In Figure 2 it can be seen that the PD behaviour at each ageing phase: PD rate, charge magnitude and phase distribution, is variable and for this reason, as can be inferred from the slopes in the damage curves, the degradation rates are also variable. During the ageing Phase C, Figure 2f, the induced damage is zero. This is because in spite of there are hot electrons during PDs, the electric field strength magnitude in the cavity is very low and those hot electrons are not effective in C–H bonds dissociation, $F_{eff}\approx 0$, see Equations (5) and (11). In Figure 2b,f the damage curves *S*_1_ and *S*_2_ are overlapped. In the case of Figure 2b, the overlapping is due to the almost deterministic PDs behaviour at the ageing Phase A, PDs magnitude and rate, independent on the polarity of the applied AC voltage. On the other hand, in the case of Figure 2f, the overlapping is due to the fact that the surfaces *S*_1_ and *S*_2_ are not damaged during the ageing Phase C. Table 2, summarizes the typical simulation results for the number of PDs per cycle, $N_{PD}$, the estimated life with Equation (6), L (h), and the mean and maximum values of the real, q (pC), and induced, $q^{\prime }$ (pC), PD charge at the five ageing phases considered. 

In Table 2, it can be seen that despite the number of PDs per cycle is the highest at phase C, there is not induced damage and on the contrary, at phase D, when the number of PDs per cycle is the lowest, the cavity surface is highly degraded. This allows inferring that the induced damage is dependent on the maximum value of the PD charge rather than on the mere PD rate. The measured number of PDs per cycle, $N_{PD\text{-}meas}$, were taken from [25] and the difference among the simulated and measured number of PDs per cycle, Diff (%), is also shown in Table 2. The maximum difference was 14.29% at the ageing Phase A, the analysis of this difference and the validation of the simulation results were presented in [29]. In Figure 2 and Table 2, it can also be seen, with exception during Phase C that the degradation rate increases with ageing. This can be explained by the fact that with ageing the cavity gas pressure is diminished and the mean free path increases, causing those electrons during PDs acquire higher kinetic energies even under electric fields of lower strength magnitude. Additionally, in Table 2, it can be seen that the cavity surface is highly degraded during Phases E and D and despite of the PD charge is higher at Phase D than at Phase E, the estimated life at Phase E is lower than that at Phase D. This allows inferring that the maximum PD charge magnitude cannot be taken as the only indicator of the ageing rate. 

The life was estimated with Equation (6) and the distributions obtained at each of the five ageing phases considered were fitted to the Weibull function using the maximum likelihood method [64]. The results of the scale, $\alpha_{L}$ (h), and shape, $\beta_{L}$, parameters, as well as the minimum and maximum values of the estimated life and the time span, $t_{span}$ (h), of each ageing phase are shown in Table 3.

The values between parentheses in Table 3 correspond to the 95% confidence intervals. As the damage in the cavity surface accumulates along the ageing phases, the time-to-breakdown under the last ageing phase can be calculated using the Miner law as follows [20]:(17)$$\sum_{j}\frac{t_{span}^{j}}{L_{j}}=1,$$
where $t_{span}^{j}$ (h) is the time span of the *j*-th ageing phase and $L_{j}$ (h) is the life estimated during the *j*-th ageing phase. Using the Equation (17) and the results for $\alpha_{L}$ presented in Table 3, the time-to-breakdown under the last ageing phase can be calculated as $t_{br}=t_{5}=$ 111.67 (106.99–116.55) h, where the values between parentheses were calculated using the confidence intervals for $\alpha_{L}$ in Table 3. From unaged conditions, the time-to-breakdown corresponds to $t_{br}=$ 1346.67 (1341.99–1351.55) h. Authors in [25], reported that tree inception was detected at ~1300 h, i.e., the time-to-breakdown calculated with the simulation results exhibit a difference of 3.59% compared with the measured one. This allows inferring that the implemented approach based on simulations is reliable and that the chemical degradation mechanism is active and predominates during the ageing phases A, B, D and E while during the ageing phase C, remains inactive. 

Figure 3 shows a superposition of the electric field strength magnitude at the cavity centre during the first period of the ageing Phase A.

In Figure 3, $E_{LAP}$ (V·m^−1^) is the Laplacian electric field strength magnitude at the cavity centre imposed by the applied voltage source, $E_{q}$ (V·m^−1^) is the electric field strength magnitude produced by the charge deployed by the PDs on the cavity surface, $E_{Inc}$ (V·m^−1^) is the PD inception magnitude and $E_{res}$ (V·m^−1^) is the resultant electric field strength at the cavity centre. Using the Equation (16) the electric field strength magnitude in the solid dielectric at r=a+5 μm, the half of the length of a unitary tree channel [59], and θ= 180° was calculated during the first period of the ageing Phase A and the result is presented in Figure 4.

In Figure 4
$E_{ch}$ (V·m^−1^) is the electric field strength magnitude in the solid dielectric and $E_{th}$ (V·m^−1^) is threshold magnitude for incepting avalanches in the microscopic channels detailed in Section 4.2. In Figure 4, it can be seen that the electric field strength magnitude in the solid dielectric close to the gas-solid interface is more than three orders of magnitude higher than both, the electric field strength at the cavity centre and the threshold magnitude for avalanches. This result allows deducing that during the first ageing phases, the surface is not sufficiently degraded for avalanches to be generated from the gas-solid interface across the solid dielectric and the chemical degradation mechanism predominates. At the final ageing phases, when the surface is highly degraded, is very likely that the physical mechanism of avalanches in the solid predominates, which can explain why the magnitude of the time-to-breakdown calculated using the chemical mechanism is slightly higher than the experimentally observed. Additionally, from the Figure 4, it can be concluded that avalanches can be generated even between, or without, PD in the cavity and as the resultant electric field is much higher than the threshold and the degradation process is very fast. The degradation rate associated with the physical mechanism will be dependent on the time dynamics of the surface-charge-decay process.

Despite the good results, the values obtained using the simulations based approach implemented here must be regarded as merely quantitative inferences because the effect of the other degradation mechanism in conjunction with the chemical, e.g., physical, UV radiation, autoxidation, etc., should be considered as well as the simultaneous variations in the cavity morphology as described in Section 3. 

### 5.2. Case Study 2, PDs Induced Degradation at Different Applied Voltage Magnitudes

In the previous case study, it could be verified that the implemented approach allows inferring quantitatively the ageing rates and that the calculated time-to-breakdown is close to the measured one. For those reasons, the same approach is used for evaluating the effect of the applied voltage magnitude on the ageing rate. The second case study was presented in [27], it corresponds to a spherical cavity of radius a=0.7 mm filled with air immersed in a solid dielectric bulk of epoxy resin between two parallel plates, D=2 mm, see Figure 1. The parameters of the media are presented in Table 1 and in [27]. The stochastic model presented in [27] was implemented in the hybrid PD-Finite Element Analysis model [29]. A 50 Hz, AC voltage source was applied to the upper electrode in the range 14 to 20 kV, while the lower electrode remained grounded. A total of 30 simulations were implemented at each voltage magnitude during 500 cycles of the AC voltage, and Figure 5 shows typical simulation results for the PRPD pattern and the accumulated degradation function curve. 

In Figure 5, it can be seen that the ageing rate, the slope of the damage function curves, increases with the magnitude of the applied voltage. Table 4 summarizes the typical simulation results for $N_{PD}$, ${q^{\prime }}_{mean}$ and L at each applied voltage magnitude. Additionally, the typical distributions of the induced PD charge obtained at each voltage amplitude were fitted to the Weibull function and the obtained scale parameter, $\alpha_{q^{\prime }}$ (pC), is also shown in Table 4**.**


The Figure 6, shows a comparison among the accumulated damage, 1/L (p.u.), and the product $N_{PD}\alpha_{q^{\prime }}$ (p.u.), calculated using the values presented in Table 4 normalized to their relative maximum values. The trend line of the accumulated damage curve, as well as its equation, calculated using the least square method, are also presented as “linear” in Figure 6. 

It can be seen that the accumulated damage, 1/L, and the product $N_{PD}\alpha_{q^{\prime }}$ increase with the applied voltage at an approximate similar rate. In [26], it was proposed the product $N_{PD}\alpha_{q^{\prime }}$ as an indicator of the energy dissipated by the PDs during each voltage cycle. In Figure 6, it can be seen that the magnitude of this product increases as the life diminishes which demonstrates that this is a good indicator of the degradation rate even under different applied voltages. The life was estimated using the Equation (6) for each of the 30 simulations at each applied voltage and the obtained distributions were fitted to the Weibull function. The results of the scale and shape parameters, as well as the minimum and maximum values of the estimated life, are presented in Table 5. 

From results in Table 5, taking into account that the confidence intervals for $\alpha_{L}$ do not overlap, and damage curves in Figure 5, it can be inferred that the degradation rate increases with the applied voltage magnitude. Additionally, in Figure 6 it can be seen that the increment in the degradation rate with the voltage magnitude is approximately linear. 

### 5.3. Case Study 3, PDs Induced Degradation at Different Applied Voltage Frequencies

The third case study is used for evaluating the effect of the frequency of the applied voltage on the ageing rate. It corresponds to a spherical cavity of radius a=0.775 mm filled with air, immersed in a solid dielectric bulk of epoxy resin of thickness D=2 mm. The parameters of the media and the test arrangement are presented in Table 1 and in [28]. A 14 kV, AC voltage source was applied to the upper electrode in the range 1–50 Hz, while the lower electrode is grounded. At each frequency, 30 simulations were implemented during 300 cycles of the AC voltage and Figure 7 shows typical simulation results for the PRPD pattern and the accumulated degradation function curve.

In Figure 7, it can be seen that the degradation rate increases with the frequency of the applied AC voltage. Table 6 summarizes the typical simulation results for $N_{PD}$, ${q^{\prime }}_{mean}$ and L at each of the applied frequencies. Additionally, it is also shown the scale parameter, $\alpha_{q^{\prime }}$, found after the fitting of the Weibull function to the typical induced PD charge simulation results.

Figure 8 shows a comparison among the accumulated damage, 1/L (p.u.), and the product $N_{PD}\alpha_{q^{\prime }}$ (p.u.), calculated using the values presented in Table 6 and normalized to their relative maximum values.

In Figure 8, it can be seen that the product $N_{PD}\alpha_{q^{\prime }}$ increases with frequency and increases as the life diminishes which allows to infer that it is a good indicator of the ageing rate, even at different frequencies. The trend line of the accumulated damage curve as well as its equation, calculated using the least square method, are also presented as “linear” in Figure 8. In this figure, it can be seen that the increment in the degradation rate with frequency seems to be higher for frequencies below 20 Hz than that above this frequency value. Besides, equation of the trend line can be rewritten as: y = 0.046x − 0.0098, for frequencies below or equal to 20 Hz, and y = 0.0026x + 0.87, for frequencies higher than 20 Hz. For each of the 30 simulations at the 5 different frequencies considered, the life was estimated using the Equation (6) and the obtained distributions were fitted to the Weibull function. The values of the scale and shape parameters, as well as the maximum and minimum values of the estimated life are presented in Table 7. 

From Table 7, taking into account that the confidence intervals do not overlap, and the damage curves in Figure 7 it can be inferred that the degradation rate increases with the frequency of the applied voltage. However, in Figure 8 it can be seen that the increment in the degradation rate is linear in two different frequency ranges i.e., between 1 Hz and 20 Hz and between 20 and 50 Hz, with the highest increment trend in the first range. This can be explained by the fact that when the frequency diminishes, the PD charge increases, see Table 6, increasing the number of electrons for collisions. In addition, the time lag between consecutive PDs also increases and the electric field strength magnitude during PDs will be higher increasing the effectiveness of electrons in produce bond dissociation by each PD, see Equation (11).

### 5.4. General Discussion

The implemented approach based on simulations and the proposed damage function, Equation (5), allows evaluating quantitatively the degradation induced by PDs activity inside solid dielectric polymers. The degradation induced by PDs in cavities is mainly driven by chemical mechanisms activated by hot electrons during PDs. It was found that the cavity surface was not degraded if the electric field strength magnitude in the cavity, determined from Equation (11), is below 2.28 kV·mm^−1^, when the hot electrons are not effective in DEA, $F_{eff}\to 0$. 

It was deduced that during the first ageing phases the chemical degradation mechanism predominated over the physical one because the surface is not sufficiently degraded for allowing electron avalanches across the solid dielectric to be started. On the other hand, at the last ageing phases, the physical mechanism predominates and the degradation is faster than under the chemical mechanism due to the enhancement of the local electric field strength at the charge injection points by the space charge. Finally, it can also be deduced that, supposing the existence of ionized gas in the cavity before and after PDs, avalanches can be generated between PDs due to the dynamics of the space charge decay. More research is needed for corroborate this.

The time-to-breakdown calculated with simulation results for a case study is close to the measured one and allows inferring that the implemented approach based on simulations is reliable. Additionally, the simulation-based approach was used for evaluating the effect of the magnitude and frequency of the applied voltage on the degradation rate and it can be concluded that the degradation rate increases, life diminishes, in a proportion approximately linear with the magnitude of the applied voltage. Similarly, the degradation rate increases, life diminishes, linearly with the frequency of the applied voltage, but at a steeper rate below 20 Hz than over this frequency value. 

The value of the product $N_{PD}\alpha_{q^{\prime }}$ is indicative of the degradation rate and the simulations results at different frequencies and magnitudes of the applied voltage allows to infer that it is a good indicator of the degradation rate even under different electrical stress conditions and can be used with online measurements for implementing prognosis tools. 

As the microscopic model employed is based on electron scattering theories, a precise characterization of the streamers landing pattern is required, so it is necessary to accurately model the PD propagation trajectory from different inception points along the cavity surface. Improvement to the calculation results can be obtained using the same simulation based approach for evaluating the PDs induced degradation presented in this study with PD plasma models instead of PD Finite Element Analysis (FEA) models. On the other hand, in the considered microscopic model it is considered that C–C single bonds are transformed into double bonds plus H^−^ ions. Under real conditions, the damage growth rate is increased due to oxidation, cross-linking between different chains, graphitization and mass reduction by C–C bonds dissociation. Due to the above, the results obtained with this method should only be considered a rough estimate of the time-to-breakdown.

The simulation-based approach proposed in this paper can be used in practical applications for implementing prognosis tools in conjunction with Artificial Intelligence (AI) methods applied to PDs diagnosis. Machine learning and AI allow to infer the location and size of cavities inside solid dielectrics as well as the cavity surface conditions [65]. Then, the simulation-based approach proposed here can be used for inferring the time-to-breakdown under the conditions previously inferred using AI.

## 6. Conclusions

In this paper, a brief phenomenological description of the degradation of solid polymeric materials induced by PDs activity in spherical cavities was presented. Then, the accumulated degradation and expected time-to-breakdown or remaining life of the insulation system were calculated using a proposed simulation based approach and a novel damage function. Simulation results showed good agreement when compared with measurements reported by other authors. Finally, the proposed simulation based approach was applied for studying the effects on the degradation rate due to variations on the applied voltage frequency and magnitude. Analytical expressions were determined for the accumulated damage as a function of the applied voltage frequency and magnitude. 

## Figures and Tables

**Figure 1 polymers-13-00324-f001:**
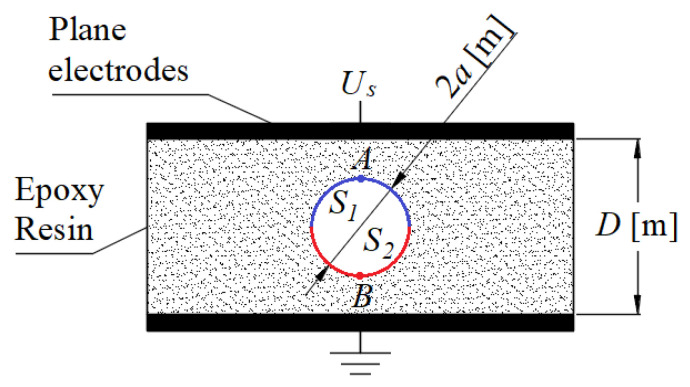
Geometry of the case studies. Material specifications, applied voltage and dimension magnitudes presented in Table 1.

**Figure 2 polymers-13-00324-f002:**
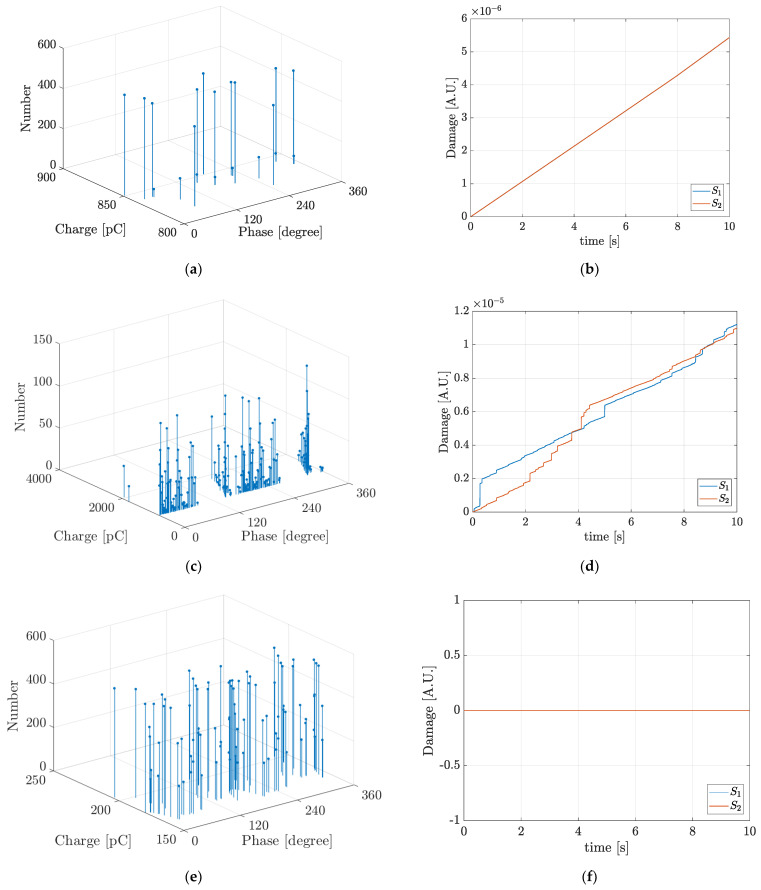
Typical simulation results of Phase-Resolved PD PRPD pattern and accumulated damage function for the five ageing phases in the first case study: (**a**) and (**b**) Phase A; (**c**) and (**d**) Phase B; (**e**) and (**f**) Phase C; (**g**) and (**h**) Phase D and (**i**) and (**j**) Phase E.

**Figure 3 polymers-13-00324-f003:**
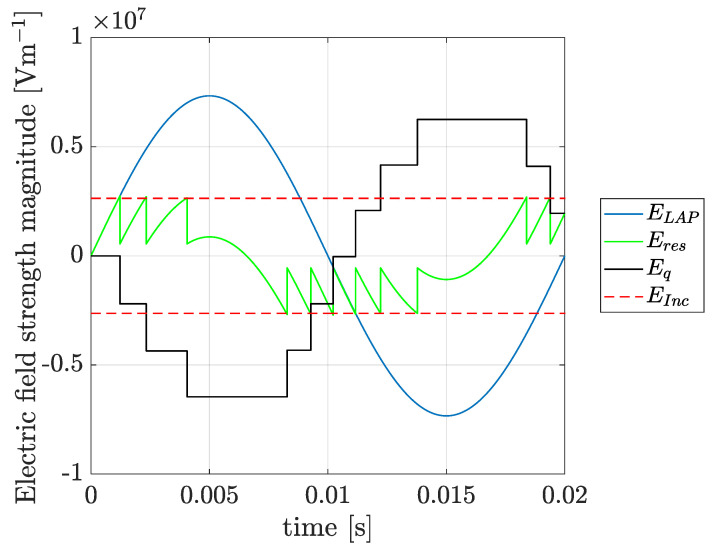
Superposition of the electric field strength magnitudes during the first cycle of the ageing Phase A.

**Figure 4 polymers-13-00324-f004:**
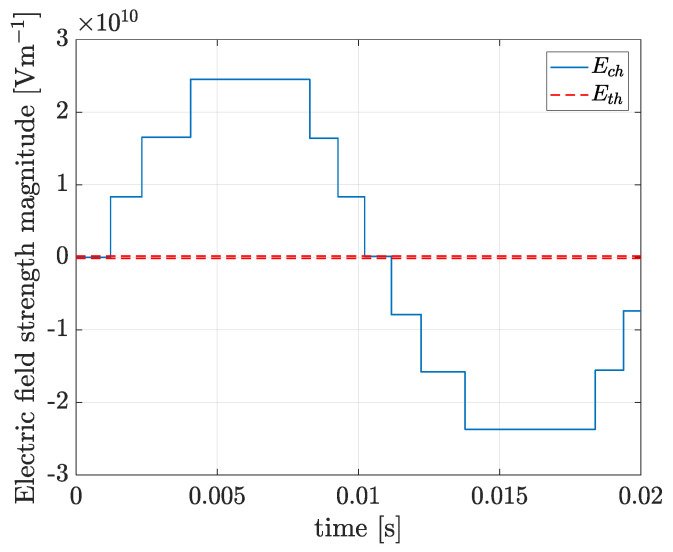
Electric field strength magnitude in the solid dielectric at *r* = *a* + 5 μm and *θ* = 180°, during the first cycle of the ageing Phase A.

**Figure 5 polymers-13-00324-f005:**
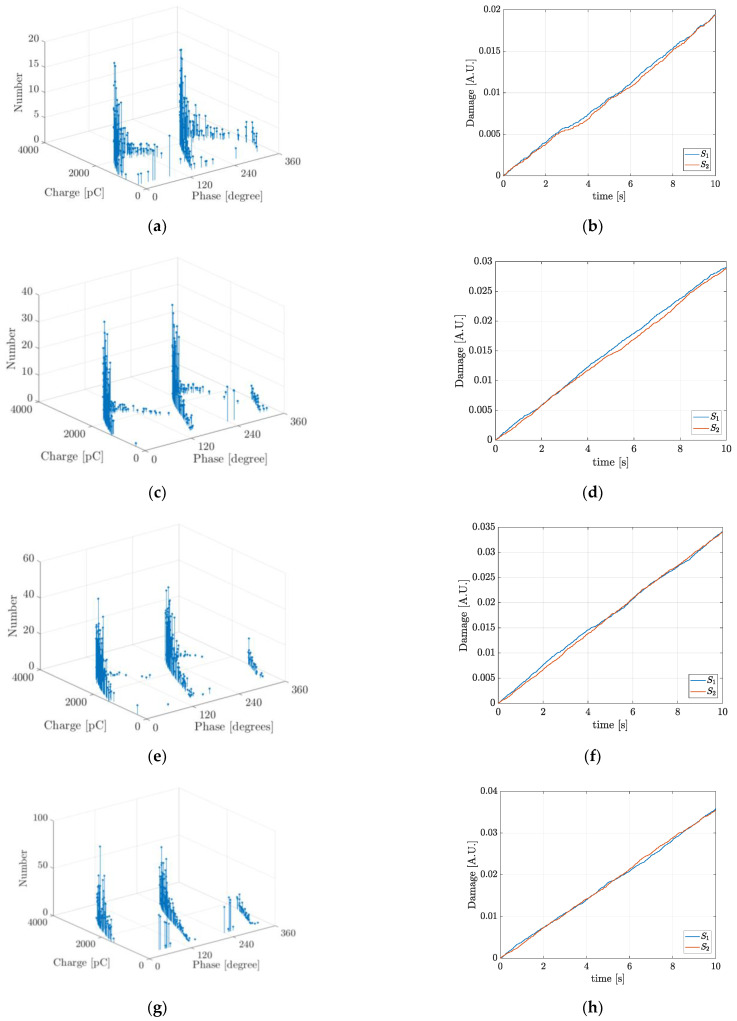
Typical simulation results of PRPD pattern and accumulated damage function for the four applied voltage magnitudes in the second case study: (**a**) and (**b**) 14 kV; (**c**) and (**d**) 16 kV; (**e**) and (**f**) 18 kV; (**g**) and (**h**) 20 kV.

**Figure 6 polymers-13-00324-f006:**
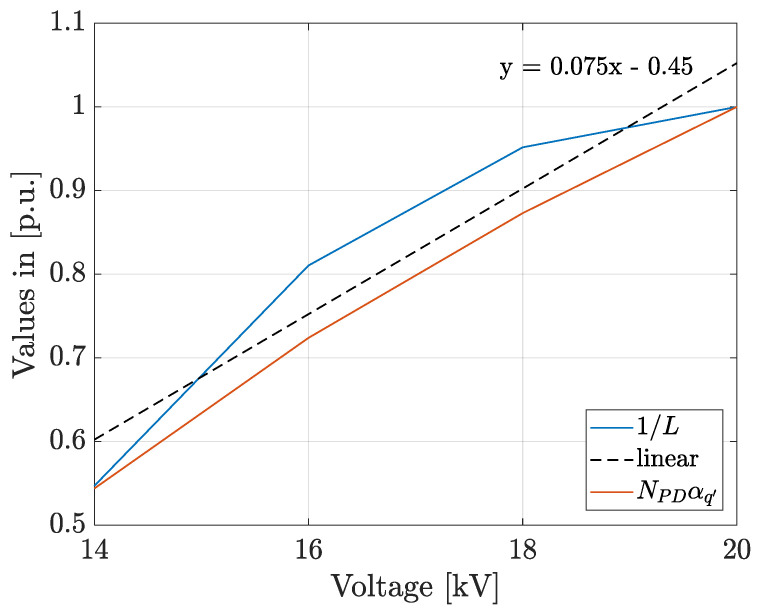
Comparison of the accumulated damage and the ageing rate function, $N_{PD}\alpha_{q^{\prime }}$, for different voltage magnitudes.

**Figure 7 polymers-13-00324-f007:**
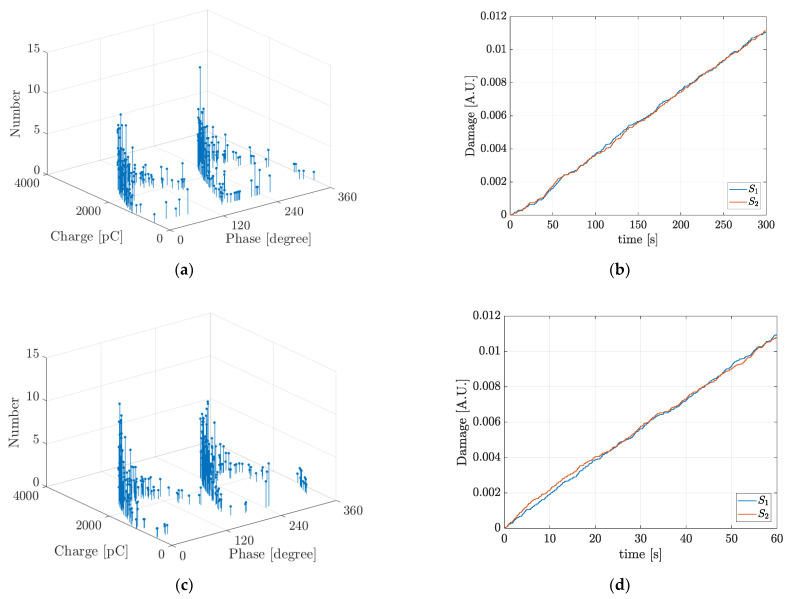
Typical simulation results of PRPD pattern and accumulated damage function for the five applied voltage frequencies in the third case study: (**a**) and (**b**) 1 Hz; (**c**) and (**d**) 5 Hz; (**e**) and (**f**) 10 Hz; (**g**) and (**h**) 20 Hz and (**i**) and (**j**) 50 Hz.

**Figure 8 polymers-13-00324-f008:**
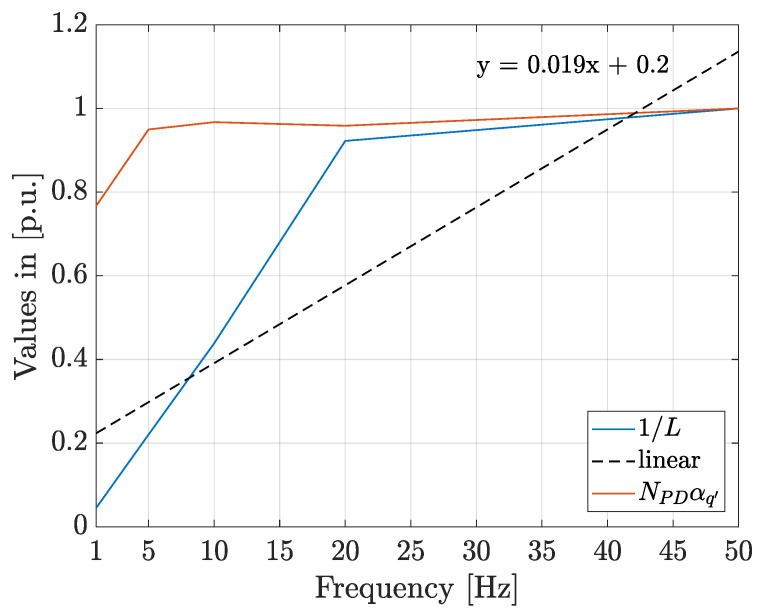
Comparison of the accumulated damage and ageing rate function, $N_{PD}\alpha_{q^{\prime }}$, for different voltage frequencies.

**Table 1 polymers-13-00324-t001:** Description of the materials and applied electric stress in the case studies.

Case Study	Material	Cavity Gas	Cavity Radius, *a* [mm]	Electric Stress
1	Epoxy resin, EP 100 (Araldite D/HY 956)	Air (2–65 kPa, 300 K)	1.25	AC, 50 Hz, 19.25 kV, *D* = 3.50 mm
2	Epoxy resin, Araldite Rapid	Air (77 kPa, 293 K)	0.70	AC, 50 Hz, 14 kV–20 kV, *D* = 2 mm
3	Epoxy resin, Araldite Rapid	Air (101 kPa, 293 K)	0.78	AC, 1 Hz–50 Hz, 14 kV, *D* = 2 mm

**Table 2 polymers-13-00324-t002:** Summary of the typical simulation results at the five ageing phases considered.

Ageing Phase	A	B	C	D	E
$N_{PD}$	11.99	11.01	64	0.06	5.15
$N_{PD\text{-}meas}$	10–14	~11.60	~60	0.06	~6
Diff (%)	14.29	5.05	6.67	0	14.23
$q_{mean}$ (pC)	834.09	848.23	177.14	2978.90	1407.60
$q_{\max}$ (pC)	850.91	2050.70	205.18	5660.90	3859.20
${q^{\prime }}_{mean}$ (pC)	529.58	538.56	112.47	1891.40	893.72
${q^{\prime }}_{\max}$ (pC)	540.26	1302	130.30	3594.20	2450.30
L (h)	3.83 × 10^4^	1.86 × 10^4^	Inf.	1433.75	184.63

**Table 3 polymers-13-00324-t003:** Parameters of the Weibull function fitted to the life estimations from simulation results at each ageing phase as well as their maximum and minimum values.

Ageing Phase	A	B	C	D	E
$t_{span}$ (h)	0.17	35	150	1049.83	50
$\alpha_{L}$ (h)	3.83 × 10^4^ (3.83 × 10^4^–3.83 × 10^4^)	2.06 × 10^4^ (1.98 × 10^4^–2.15 × 10^4^)	Inf.	2082.86 (1760.11–2464.83)	225.93 (216.46–235.81)
$\beta_{L}$	Inf.	8.89 (6.81–11.62)	—	2.26 (1.72–2.95)	8.88 (6.85–11.53)
$L_{\max}$ (h)	3.83 × 10^4^	2.47 × 10^4^	Inf.	3566.67	273.18
$L_{\min}$ (h)	3.83 × 10^4^	1.38 × 10^4^	Inf.	708.78	179.99

**Table 4 polymers-13-00324-t004:** Summary of the typical simulation results for each applied voltage magnitude in the second case study.

Applied Voltage Magnitude (kV)	14	16	18	20
$N_{PD}$	2.58	4.15	5.66	7.27
${q^{\prime }}_{mean}$ (pC)	768.38	678.11	594.85	523.46
$\alpha_{q^{\prime }}$ (pC)	876.89	726.60	642.64	573.07
L (h)	10.62	7.17	6.11	5.81

**Table 5 polymers-13-00324-t005:** Parameters of the Weibull function fitted to the life estimations from simulation results at each applied voltage magnitude and their maximum and minimum values.

Applied Voltage Magnitude (kV)	14	16	18	20
$\alpha_{L}$ (h)	11.23 (11.07–11.40)	7.16 (7.10–7.22)	5.98 (5.94–6.02)	5.60 (5.55–5.65)
$\beta_{L}$	25.39 (19.41–33.21)	45.35 (35.12–58.56)	53.53 (40.64–70.52)	46.58 (35.95–60.37)
$L_{\min}$ (h)	10.13	6.69	5.63	5.36
$L_{\max}$ (h)	11.92	7.45	6.11	5.81

**Table 6 polymers-13-00324-t006:** Summary of the typical simulation results for each applied voltage frequency in the third case study.

Applied Voltage Frequency (Hz)	1	5	10	20	50
$N_{PD}$	2.17	2.83	3.20	3.49	5.26
${q^{\prime }}_{mean}$ (pC)	1296.70	1198.70	1082.16	1066.36	753.38
$\alpha_{q^{\prime }}$ (pC)	1279.20	1214.50	1092.50	992.86	687.18
L (h)	561.42	114.36	57.49	27.33	25.20

**Table 7 polymers-13-00324-t007:** Parameters of the Weibull function fitted to the life estimations from results at each applied voltage frequency and their maximum and minimum values.

Applied Voltage Frequency (Hz)	1	5	10	20	50
$\alpha_{L}$ (h)	610.44 (599.56–621.56)	113.57 (111.16–116.04)	54.03 (53.29–54.79)	28.07 (27.73–28.41)	24.68 (24.34–25.02)
$\beta_{L}$	21.08 (16.14–27.53)	17.72 (13.78–22.79)	27.53 (21.50–35.24)	31.29 (23.94–40.89)	27.46 (20.99–35.92)
$L_{\min}$ (h)	538.83	100.59	50.53	26.03	21.78
$L_{\max}$ (h)	658.69	124.50	57.58	29.50	26.18

## Data Availability

Not Applicable.

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
