# Peer review of "A Study on the Life Estimation and Cavity Surface Degradation Due to Partial Discharges in Spherical Cavities within Solid Polymeric Dielectrics Using a Simulation Based Approach"

_polymers, 2021, doi:10.3390/polym13030324_

Round 1

Reviewer 1 Report

Manuscript ID: polymers‐1071639
Johnatan M. Rodríguez‐Serna and Ricardo Albarracín‐Sánchez
A study on the life estimation and cavity surface degradation due to partial discharges in spherical
cavities within solid polymeric dielectrics using a simulation based approach.
The paper is correctly written and the research results are properly discussed and justified. However,
this paper needs to be improved as detailed in the comments below.
Detailed comments referring to the paper:
1. Sections Abstract and Introduction ‐ please indicate in more detail the novelty of the paper
against the background of the literature on the subject.
2. Please emphasize the possibilities of practical application of the methods that are developed in
the paper.
3. Please write the measurement units in square brackets instead of round ones ‐ e.g. line 306.
Analogous comment applies to all Figures and Tables.
4. Line 16 ‐ please support this sentence by the relevant Reference/References.
5. Tables 2‐6: what is the reason for the different number of significant digits?
6. Conclusions are too detailed. This section should provide a general summary of the paper.

Author Response

Revisions Letter

Dear Reviewer,

We are writing to thank you for your response and your effort in reading and revising in detail our paper. You may find below your comments to the revision of the paper polymers-1071639 entitled “A study on the life estimation and cavity surface degradation due to partial discharges in spherical cavities within solid polymeric dielectrics using a simulation based approach”, and our answers.

We are also sending, separately, a new version of the paper with all changes highlighted using the "Track Changes" function in Microsoft Word. If you might have further questions, we will be very glad to answer them.

General comment:

The paper is correctly written and the research results are properly discussed and justified.

Answer

Dear reviewer, we would like to thank you for your comment.

Comment 1:

Sections Abstract and Introduction ‐ please indicate in more detail the novelty of the paper

against the background of the literature on the subject.

Answer

The following paragraph was added between lines 78-87 in the introduction section:

In this paper, it is presented a study on the degradation of polymeric materials induced by PDs in spherical cavities inside solid dielectrics implementing an innovative simulation based approach that uses a novel damage function and allows to quantitatively inferring the time-to-breakdown of the insulation system. Additionally, based on simulation results, the effectiveness as well as the period of predominance above each other of the chemical and physical degradation mechanisms are discussed. On the other hand, the simulation-based approach is used for numerically evaluating the effect of the applied voltage frequency and magnitude on the degradation rate and analytical expressions for the trend lines of the accumulated damage or ageing as a function of voltage magnitude and frequency were determined.    

Comment 2:

Please emphasize the possibilities of practical application of the methods that are developed in

the paper.

Answer

The following paragraph was added between lines 661-666 in section 5.4:

The simulation-based approach proposed in this paper can be used in practical applications for implementing prognosis tools in conjunction with Artificial Intelligence (AI) methods applied to PDs diagnosis. Machine learning and AI allow to infer the location and size of cavities inside solid dielectrics as well as the cavity surface conditions [65]. Then, the simulation-based approach proposed here can be used for inferring the time-to-breakdown under the conditions previously inferred using AI.

Comment 3:

Please write the measurement units in square brackets instead of round ones ‐ e.g. line 306.

Analogous comment applies to all Figures and Tables.

Answer

We have rewritten the measurement units along all the paper, including figures and tables, between square brackets.

Comment 4:

Line 16 ‐ please support this sentence by the relevant Reference/References.

Answer

In order not to introduce references in the abstract, we added the following sentence with references between lines 53-55 in the introduction section, directly related to the sentence in line 16 in the abstract (now line 15):

As PDs activate the fastest degradation mechanism in polymeric solid dielectrics due to chemical and physical deterioration mechanisms activated by the charge carriers, UV radiation and local temperature rising during PDs activity [8,14], …

Comment 5:

Tables 2‐6: what is the reason for the different number of significant digits?

Answer

We have rewritten all values in Tables 2-6 rounding to two decimal places and used scientific notation for numbers equal or greater than 1x104.

Comment 6:

Conclusions are too detailed. This section should provide a general summary of the paper.

Answer

We have added the following paragraph between lines 668-677 in the conclusions section:

In this paper, a brief phenomenological description of the degradation of solid polymeric materials induced by PDs activity in spherical cavities was presented. Then, the accumulated degradation and expected time-to-breakdown or remaining life of the insulation system were calculated using a proposed simulation based approach and a novel damage function. Simulation results showed good agreement when compared with measurements reported by other authors. Finally, the proposed simulation based approach was applied for studying the effects on the degradation rate due to variations on the applied voltage frequency and magnitude. Analytical expressions were determined for the accumulated damage as a function of the applied voltage frequency and magnitude.    

On the other hand, we moved the conclusions in the previous version of the paper to a new sub-section in section 5 between lines 621 and 660, titled 5.4 General Discussion.

Best regards

Reviewer 2 Report

The manuscript reports the study of the life estimation and cavity surface degradation due to partial discharges in spherical cavities within epoxy resin using a simulation based approach. There are several comments here:

  1. Please sketch the discharge model. The applied electrical field and spherical cavity are orthogonal to each other. Why have discharge path on the spherical cavity and damage happens?
  2. In Figs 2(b) and 2(f), they shown two straight lines. Are curves S1 and S2 overlap? Please check.

Therefore, I recommend it as major revision to publish.

Author Response

Revisions Letter

Dear Reviewer,

We are writing to thank you for your response and your effort in reading and revising in detail our paper. You may find below your comments to the revision of the paper polymers-1071639 entitled “A study on the life estimation and cavity surface degradation due to partial discharges in spherical cavities within solid polymeric dielectrics using a simulation based approach”, and our answers.

We are also sending, separately, a new version of the paper with all changes highlighted using the "Track Changes" function in Microsoft Word. If you might have further questions, we will be very glad to answer them.

General comment:

The manuscript reports the study of the life estimation and cavity surface degradation due to partial discharges in spherical cavities within epoxy resin using a simulation based approach.

Answer

Dear reviewer, we would like to thank you for your comment.

Comment 1:

Please sketch the discharge model. The applied electrical field and spherical cavity are orthogonal to each other. Why have discharge path on the spherical cavity and damage happens?

Answer

The following paragraphs were added between lines 365-400, in section 5.

The hybrid PD-Finite Element Analysis model can be summarized as in the following steps [29]:

  1. The parameters of media, geometrical constants, boundaries and subdomains are defined;
  2. For each time-step the electric field strength is calculated using a Finite Element Analysis solver at all domains;
  3. For each time-step the inception criteria are electron existence and minimal electric field strength magnitude and both are verified;
  4. If the inception criteria are fulfilled, the cavity conductivity is increased and the time-step is diminished. The high-conductivity condition is maintained until the electric field strength magnitude is lower or equal to an extinction magnitude.
  5. If inception criteria are not fulfilled, the charge distribution on the cavity surface deployed by previous PDs is calculated as a function of the electric field strength inside the cavity, the time and the cavity surface conductivity.
  6. The induced charge is calculated evaluating boundary conditions at electrodes and parameters of media are reset for the following time-step. Calculations for the following time-step are executed as in step 2 until the required simulation time is reached.

In the hybrid PD-Finite Element Analysis model used here, as in the conductance and plasma models [30], the initiation and ending locations of the PDs are considered fixed specific points on the inner cavity surface, the centre of surfaces S1, A, and S2, B, Figure 1. This is because the electric field strength magnitude is the greatest at those points, which increases the probability of first electrons emission for incepting PDs [61]. Additionally, it is considered that once the PDs are incepted at points A or B, they propagate along the cavity symmetrical axis. This is because charges are accelerated by the electric field in the direction of the greatest electric field strength magnitude, until the opposite surface is reached at points B or A, where the charges left by PDs produce a field that opposes to the externally applied, quenching the PDs processes [62].

As the charges deployed by PDs are initially concentrated close to points A and B, Figure 1, where the streamer impinges the cavity surface and the electric field strength magnitude and the energy distribution of electrons are the greatest [14,61], it is assumed that the damage induced by hot electrons is accumulated at those points on the inner cavity surface. A precise evaluation of the real affected area requires the precise determination of the initiation and ending locations of PDs and the consequent damage accumulation at those locations. Plasma models can give a good alternative for modelling the precise PD landing pattern [30].

Comment 2:

In Figs 2(b) and 2(f), they shown two straight lines. Are curves S1 and S2 overlap? Please check.

Answer

In Figures 2b and 2f, the curves are overlapped. We have added the following text between lines 433-438 in the paragraph below Figure 2:

In Figures 2b and 2f the damage curves S1 and S2 are overlapped. In the case of Figure 2b, the overlapping is due to the almost deterministic PDs behaviour at the ageing Phase A, PDs magnitude and rate, independent on the polarity of the applied AC voltage. On the other hand, in the case of Figure 2f, the overlapping is due to the fact that the surfaces S1 and S2 are not damaged during the ageing Phase C.

Best regards

Round 2

Reviewer 2 Report

The manuscript was revised well. Therefore, in my opinion, the article could be accepted to publish.